# An Empirical Bayes Approach to Optimizing Machine Learning Algorithms

**James McInerney**
Spotify Research
45 W 18th St, 7th Floor
New York, NY 10011
`jamesm@spotify.com`

## Abstract

There is rapidly growing interest in using Bayesian optimization to tune model and inference hyperparameters for machine learning algorithms that take a long time to run. For example, Spearmint is a popular software package for selecting the optimal number of layers and learning rate in neural networks. But given that there is uncertainty about which hyperparameters give the best predictive performance, and given that fitting a model for each choice of hyperparameters is costly, it is arguably wasteful to "throw away" all but the best result, as per Bayesian optimization. A related issue is the danger of overfitting the validation data when optimizing many hyperparameters. In this paper, we consider an alternative approach that uses more samples from the hyperparameter selection procedure to average over the uncertainty in model hyperparameters. The resulting approach, empirical Bayes for hyperparameter averaging (EB-Hyp) predicts held-out data better than Bayesian optimization in two experiments on latent Dirichlet allocation and deep latent Gaussian models. EB-Hyp suggests a simpler approach to evaluating and deploying machine learning algorithms that does not require a separate validation data set and hyperparameter selection procedure.

## 1 Introduction

There is rapidly growing interest in using Bayesian optimization (BayesOpt) to tune model and inference hyperparameters for machine learning algorithms that take a long time to run (Snoek *et al.*, 2012). Tuning algorithms by grid search is a time consuming task. Tuning by hand is also time consuming and requires trial, error, and expert knowledge of the model. To capture this knowledge, BayesOpt uses a performance model (usually a Gaussian process) as a guide to regions of hyperparameter space that perform well. BayesOpt balances exploration and exploitation to decide which hyperparameter to evaluate next in an iterative procedure.

BayesOpt for machine learning algorithms is a form of model selection in which some objective, such as predictive likelihood or root mean squared error, is optimized with respect to hyperparameters $\eta$. Thus, it is an empirical Bayesian procedure where the marginal likelihood is replaced by a proxy objective. Empirical Bayes optimizes the marginal likelihood of data set $X$ (a summary of symbols is provided in Table 1),

$$\hat{\eta} \coloneqq \arg \max_{\eta} \mathbb{E}_{p(\theta \mid \eta)}[p(X \mid \theta)], \tag{1}$$

then uses $p(\theta \mid X, \hat{\eta})$ as the posterior distribution over the unknown model parameters $\theta$ (Carlin and Louis, 2000). Empirical Bayes is applied in different ways, e.g., gradient-based optimization of Gaussian process kernel parameters, optimization of hyperparameters to conjugate priors in variational inference. What is special about BayesOpt is that it performs empirical Bayes in a way

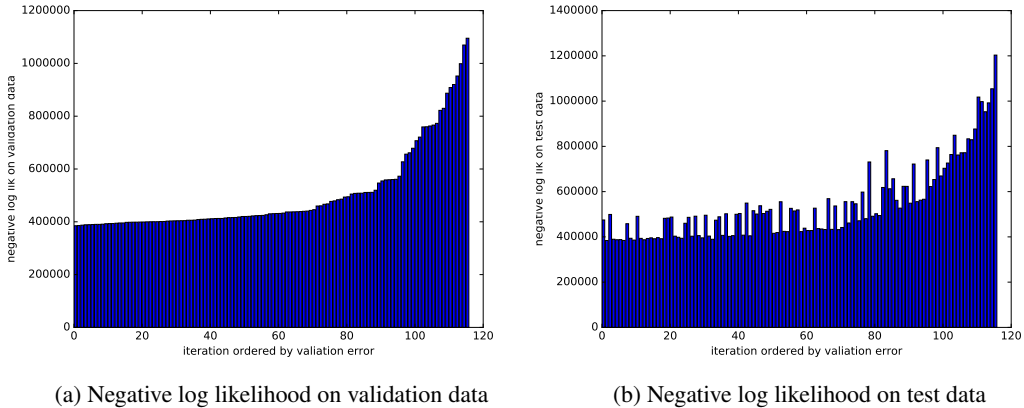

(a) Negative log likelihood on validation data  (b) Negative log likelihood on test data

Figure 1: Performance in negative logarithm of the predictive likelihood for the validation data (left plot) and test data (right plot) ordered by validation error. Each iteration represents a different hyperparameter setting.

Table 1: Summary of Symbols

| Symbol | Meaning |
|---|---|
| $\theta$ | the model parameters |
| $\eta$ | the hyperparameters |
| $\lambda$ | the hyper-hyperparameters |
| $\hat{\eta}$ | the hyperparameters fit by empirical Bayes |
| $\hat{\lambda}$ | the hyper-hyperparameters fit by empirical Bayes |
| $X$ | the dataset |
| $X^*$ | unseen data |

that requires calculating the posterior $p(\theta \mid X, \eta^{(s)})$ for *each* member in a sequence $1, \dots, S$ of candidate hyperparameters $\eta^{(1)}, \eta^{(2)}, \dots, \eta^{(S)}$. Often these posteriors are approximate, such as a point estimate, a Monte Carlo estimate, or a variational approximation. Nonetheless, these operations are usually expensive to compute.

Therefore, what is surprising about BayesOpt for approximate inference is that it disregards most of the computed posteriors and keeps only the posterior $p(\theta \mid X, \hat{\eta})$ that optimizes the marginal likelihood. It is surprising because the intermediate posteriors have something to say about the data, even if they condition on hyperparameter configurations that do not maximize the marginal likelihood. In other words, when we harbour uncertainty about $\eta$, should we be more Bayesian? We argue for this approach, especially if one believes there is a danger of overfitting $\eta$ on the validation set, which is especially the case as the dimensionality of the hyperparameters grows. As an illustrative example, Figure 1 shows the predictive performance of a set of 115 posteriors (each corresponding to a different hyperparameter) of latent Dirichlet allocation on validation data and testing data. Overfitting validation means that the single best posterior would not be selected as the final answer in BayesOpt.

Bayes empirical Bayes (Carlin and Louis, 2000) extends the empirical Bayes paradigm by introducing a family of hyperpriors $p(\eta \mid \lambda)$ indexed by $\lambda$ and calculates the posterior over the model parameters by integrating,

$$p(\theta \mid X, \lambda) = \mathbb{E}_{p(\eta \mid X, \lambda)}[p(\theta \mid X, \eta)]. \tag{2}$$

This leads to the question of how to select the hyper-hyperparameter $\lambda$. A natural answer is a hierarchical empirical Bayes approach where $\lambda$ is maximized[1],

$$\hat{\lambda} = \arg \max_{\lambda} \mathbb{E}_{p(\eta \mid \lambda)} \mathbb{E}_{p(\theta \mid \eta)}[p(X \mid \theta, \eta)], \tag{3}$$

and $p(\theta \mid X, \hat{\lambda})$ is used as the posterior. Comparing Eq. 3 to Eq. 1 highlights that we are adding an extra layer of marginalization that can be exploited with the intermediate posteriors in hand. Note the distinction between marginalizing the hyperparameters to the model vs. hyperparameters to the Gaussian process of model performance. Eq. 3 describes the former; the latter is already a staple of BayesOpt (Osborne, 2010).

In this paper, we present *empirical Bayes for hyperparameter averaging* (EB-Hyp), an extension to BayesOpt that makes use of this hierarchical approach to incorporate the intermediate posteriors in an approximate predictive distribution over unseen data $X^*$.

**The Train-Marginalize-Test Pipeline**   EB-Hyp is an alternative procedure for evaluating and deploying machine learning algorithms that reduces the need for a separate validation data set.

Validation data is typically used to avoid overfitting. Overfitting is a danger in selecting both parameters and hyperparameters. The state of the art provides sophisticated ways of regularizing or marginalizing over parameters to avoid overfitting on training data. But there is no general method for regularizing hyperparameters and typically there is a requirement of conjugacy or continuity in order to simultaneously fit parameters and hyperparameters in the same training procedure.

Therefore, the standard practice for dealing with the hyperparameters of machine learning models and algorithms is to use a separate validation data set (Murphy, 2012). One selects the hyperparameter that results in the best performance on validation data after fitting the training data. The best hyperparameter and corresponding posterior are then applied to a held-out test data set and the resulting performance is the final estimate of the generalization performance of the entire system. This practice of separate validation has carried over to BayesOpt.

EB-Hyp avoids overfitting training data through marginalization and allows us to train, marginalize, and test without a separate validation data set. It consists of three steps:

1. **Train** a set of parameters on training data $X_{\text{train}}$, each one conditioned on a choice of hyperparameter.
2. **Marginalize** the hyperparameters out of the set of full or approximate posteriors.
3. **Test** (or Deploy) the marginal predictive distribution on test data $X_{\text{test}}$ and report the performance.

In this paper, we argue in favour of this framework as a way of simplifying the evaluation and deployment pipeline. We emphasize that the *train* step admits a broad category of posterior approximation methods for a large number of models, including maximum likelihood, maximum *a posteriori*, variational inference, or Markov chain Monte Carlo.

In summary, our contributions are the following:

- We highlight the three main shortcomings of the current prevalent approach to tuning hyperparameters of machine learning algorithms (computationally wasteful, potentially overfitting validation, added complexity of a separate validation data set) and propose a new empirical Bayes procedure, EB-Hyp, to address those issues.
- We develop an efficient algorithm to perform EB-Hyp using Monte Carlo approximation to both sample hyperparameters from the marginal posterior and to optimize over the hyper-hyperparameters.
- We apply EB-Hyp to two models and real world data sets, comparing to random search and BayesOpt, and find a significant improvement in held out predictive likelihood validating the approach and approximation in practice.

## 2   Related Work

Empirical Bayes has a long history started by Robbins (1955) with a nonparametric approach, to parametric EB (Efron and Morris, 1972) and modern applications of EB (Snoek *et al.*, 2012; Rasmussen and Williams, 2006). Our work builds on these hierarchical Bayesian approaches.

BayesOpt uses a GP to model performance of machine learning algorithms. A previous attempt at reducing the wastefulness of BayesOpt has focused on directing computational resources toward

more optimal regions of hyperparameter space (Swersky *et al.*, 2014). Another use of the GP as a performance model arises in Bayesian quadrature, which uses a GP to approximately marginalize over parameters (Osborne *et al.*, 2012). However, quadrature is computationally infeasible for forming a predictive density after marginalizing hyperparameters because that requires knowing $p(\theta \mid X, \eta)$ for the whole space of $\eta$. In contrast, the EB-Hyp approximation depends on the posterior only at the sampled points, which has already been calculated to estimate the marginals.

Finally, EB-Hyp resembles ensemble methods, such as boosting and bagging, because it is a weighted sum over posteriors. Boosting trains models on data reweighted to emphasize errors from previous models (Freund *et al.*, 1999) while bagging takes an average of models trained on bootstrapped data (Breiman, 1996).

# 3 Empirical Bayes for Hyperparameter Averaging

As introduced in Section 1, EB-Hyp adds another layer in the model hierarchy with the addition of a hyperprior $p(\eta \mid \lambda)$. The Bayesian approach is to marginalize over $\eta$ but, as usual, the question of how to select the hyper-hyperparameter $\lambda$ lingers. Empirical Bayes provides a response to the selection of hyperprior in the form a maximum marginal likelihood approach (see Eq. 3). It is useful to incorporate maximization into the posterior approximation when tuning machine learning algorithms because of the small number of samples we can collect (due to the underlying assumption that the inner training procedure is expensive to run).

Our starting point is to approximate the posterior predictive distribution under EB-Hyp using Monte Carlo samples of $\eta^{(s)} \sim p(\eta \mid X, \hat{\lambda})$,

$$p(X^* \mid X) \approx \frac{1}{S} \sum_{s=1}^{S} \mathbb{E}_{p(\theta \mid X, \eta^{(s)})}[p(X^* \mid \theta, \eta^{(s)})] \tag{4}$$

for a choice of hyperprior $p(\eta \mid \lambda)$.

There are two main challenges that Eq. 4 presents. The first is that the marginal posterior $p(\eta \mid X, \hat{\lambda})$ is not readily available to sample from. We address this in Section 3.1. The second is the choice of hyperprior $p(\eta \mid \lambda)$ and how to find $\hat{\lambda}$. We describe our approach to this in Section 3.2.

## 3.1 Acquisition Strategy

The acquisition strategy describes which hyperparameter to evaluate next during tuning. A naïve way to choose evaluation point $\eta$ is to sample from the uniform distribution or the hyperprior. However, this is likely to select a number of points where $p(X \mid \eta, \lambda)$ has low density, squandering computational resources.

BayesOpt addresses this by using an acquisition function conditioned on the current performance model posterior then maximizing this function to select the next evaluation point. BayesOpt offers several choices for the acquisition function. The most prominent are expected improvement, upper confidence bound, and Thompson sampling (Brochu *et al.*, 2010; Chapelle and Li, 2011). Expected improvement and the upper confidence bound result in deterministic acquisition functions and are therefore hard to incorporate into Eq. 4, which is a Monte Carlo average. In contrast, Thompson sampling is a stochastic procedure that is competitive with the non-stochastic procedures (Chapelle and Li, 2011), so we use it as a starting point for our acquisition strategy.

Thompson sampling maintains a model of rewards for actions performed in an environment and repeats the following for iteration $s = 1, \ldots, S$:

1. Draw a simulation of rewards from the current reward posterior conditioned on the history $r^{(s)} \sim p(r \mid \{\eta^{(t)}, f^{(t)} \mid t < s\})$.

2. Choose the action that gives the maximum reward in the simulation $\eta^{(s)} = \arg\max_\eta r^{(s)}(\eta)$.

3. Observe reward $f^{(s)}$ from the environment for performing action $\eta^{(s)}$.

Thompson sampling balances exploration with exploitation because actions with large posterior means and actions with high variance are both more likely to appear as the optimal action in the sample $r^{(s)}$. However, the $\arg\max$ presents difficulties in the reweighting required to perform Bayes empirical Bayes approaches. We discuss these difficulties in more depth in Section 3.2. Furthermore, it is unclear exactly what the sample set $\{\eta^{(1)}, \ldots, \eta^{(S)}\}$ represents. This question becomes pertinent when we care about more than just the optimal hyperparameter. To address these issues, we next present a procedure that generalizes Thompson sampling when it is used for hyperparameter tuning.

**Performance Model Sampling**   Performance model sampling is based on the idea that the set of simulated rewards $r^{(s)}$ can themselves be treated as a probability distribution of hyperparameters, from which we can also draw samples. In a hyperparameter selection context, let $\tilde{p}^{(s)}(X \mid \eta) \equiv r^{(s)}$, the marginal likelihood. The procedure repeats for iterations $s = 1, \ldots, S$:

1. **draw** $\tilde{p}^{(s)}(X \mid \eta) \sim \mathcal{P}(p(X \mid \eta) \mid \{\eta^{(t)}, f_X^{(t)} \mid t < s\})$

2. **draw** $\eta^{(s)} \sim \tilde{p}^{(s)}(\eta \mid X)$

3. **evaluate** $f_X^{(s)} = \int p(X \mid \theta) p(\theta \mid \eta^{(s)}) \mathrm{d}\theta$

where $\tilde{p}^{(s)}(\eta \mid X) := Z^{-1} \tilde{p}^{(s)}(X \mid \eta) p(\eta)$         (5)

where $\mathcal{P}$ is the performance model distribution and $Z$ is the normalization constant.[2] The marginal likelihood $p(X \mid \eta^{(s)})$ may be evaluated exactly (e.g., Gaussian process marginal given kernel hyperparameters) or estimated using methods that approximate the posterior $p(\theta \mid X, \eta^{(s)})$ such as maximum likelihood estimation, Markov chain Monte Carlo sampling, or variational inference.

Thompson sampling is recovered from performance model sampling when the sample in Step 2 of Eq. 5 is replaced with the maximum *a posteriori* approximation (with a uniform prior over the bounds of the hyperparameters) to select where to obtain the next hyperparameter sample $\eta^{(s)}$. Given the effectiveness of Thompson sampling in various domains (Chapelle and Li, 2011), this is likely to work well for hyperparameter selection. Furthermore, Eq. 5 admits a broader range of acquisition strategies, the simplest being a full sample. And importantly, it allows us to consider the convergence of EB-Hyp.

The sample $\tilde{p}^{(s)}(X \mid \eta)$ of iteration $s$ from the procedure in Eq. 5 converges to the true probability density function $p(X \mid \eta)$ as $s \to \infty$ under the assumptions that $p(X \mid \eta)$ is smooth and the performance model $\mathcal{P}$ is drawn from a log Gaussian process with smooth mean and covariance over a finite input space. Consistency of the Gaussian process in one dimension has been shown for fixed Borel probability measures (Choi and Schervish, 2004). Furthermore, rates of convergence are favourable for a variety of covariance functions using the log Gaussian process for density estimation (van der Vaart and van Zanten, 2008). Performance model sampling additionally changes the sampling distribution of $\eta$ on each iteration. Simulation $\tilde{p}^{(s)}(\eta \mid X)$ from the posterior of $\mathcal{P}$ conditioned on the evaluation history has non-zero density wherever the prior $p(\eta)$ is non-zero by the definition of $\tilde{p}^{(s)}(\eta \mid X)$ in Eq. 5 and the fact that draws from a log Gaussian process are non-zero. Therefore, as $s \to \infty$, the input-output set $\{\eta^{(t)}, f_X^{(t)} \mid t < s\}$ on which $\mathcal{P}$ is conditioned will cover the input space.

It follows from the above discussion that the samples $\{\eta^{(s)} \mid s \in [1, S]\}$ from the procedure in Eq. 5 converge to the posterior distribution $p(\eta \mid X)$ as $S \to \infty$. Therefore, the sample $\tilde{p}^{(s)}(X \mid \eta)$ converges to the true pdf $p(X \mid \eta)$ as $s \to \infty$. Since $\{\eta^{(s)} \mid s \in [1, S]\}$ is sampled independently from $\{\tilde{p}^{(s)}(X \mid \eta) \mid s \in [1, S]\}$ (respectively), the set of samples therefore tends to $p(\eta \mid X)$ as $S \to \infty$.

A key limitation to the above discussion for continuous hyperparameters is the assumption that the true marginal $p(X \mid \eta)$ is smooth. This may not always be the case, for example an infinitesimal change in the learning rate for gradient descent on a non-convex objective could result in finding a completely different local optimum. This affects asymptotic convergence but discontinuities in the

1 **inputs** training data $X_{\text{train}}$ and inference algorithm $\mathcal{A} : (X, \eta) \to p(\theta \mid X, \eta)$
2 **output** predictive density $p(X^* \mid X_{\text{train}})$
3 initialize evaluation history $V = \{\}$
4 **while** $V$ *not converged* **do**
5   draw performance function from GP posterior $\tilde{p}^{(s)}(X \mid \eta) \sim \mathcal{GP}(\cdot \mid V)$
6   calculate hyperparameter posterior $\tilde{p}^{(s)}(\eta \mid X) := Z^{-1}\tilde{p}^{(s)}(X \mid \eta)p(\eta)$
7   draw next evaluation point $\eta^{(s)} := \arg\max_\eta \tilde{p}^{(s)}(\eta \mid X)$
8   run parameter inference conditioned on hyperparameter $p(\theta \mid \eta^{(s)}) := \mathcal{A}(X_{\text{train}}, \eta^{(s)})$
9   evaluate performance $f_X^{(s)} := \int p(X_{\text{train}} \mid \theta)p(\theta \mid \eta^{(s)})\mathrm{d}\theta$
10   append $(\eta^{(s)}, f_X^{(s)})$ to history $V$
11 **end**
12 find optimal $\hat{\lambda}$ using Eq. 3 (discussed in Section 3.2)
13 **return:** approximation to $p(X^* \mid X_{\text{train}})$ using Eq. 4

**Algorithm 1:** Empirical Bayes for hyperparameter averaging (EB-Hyp)

Table 2: Predictive log likelihood for latent Dirichlet allocation (LDA), 20 Newsgroup dataset

| Method | | Predictive Log Lik. (% Improvement on BayesOpt) |
|---|---|---|
| BayesOpt | with validation | -357648 (0.00%) |
| | without validation | -361661 (-1.12%) |
| EB-Hyp | with validation | -357650 (-0.00%) |
| | without validation | **-351911 (+1.60%)** |
| Random | | -2666074 (-645%) |

marginal likelihood are not likely to affect the outcome at the scale number of evaluations typical in hyperparameter tuning. Importantly, the smoothness assumption does not pose a problem to discrete hyperparameters (e.g., number of units in a hidden layer). Another limitation of performance model sampling is that it focuses on the marginal likelihood as the metric to be optimized. This is less of a restriction as it may first appear. Various performance metrics are often equivalent or approximations to a particular likelihood, e.g., mean squared error is the negative log likelihood of a Gaussian-distributed observation.

## 3.2 Weighting Strategy

Performance model sampling provides a set of hyperparameter samples, each with a performance $f_X^{(s)}$ and a computed posterior $p(\theta \mid X, \eta^{(s)})$. These three elements can be combined in a Monte Carlo average to provide a prediction over unseen data or a mean parameter value.

Following from Section 3.1, the samples of $\eta$ from Eq. 5 converge to the distribution of $p(\eta \mid X, \lambda)$. A standard Bayesian treatment of the hierarchical model requires selecting a fixed $\lambda$, equivalent to a predetermined weighted or unweighted average of the models of a BayesOpt run. However, we found that fixing $\lambda$ is not competitive with approaches to hyperparameter tuning that involve some maximization. This is likely to arise from the small number of samples collected during tuning (recall that collecting more samples involves new entire runs of parameter training and is usually computationally expensive).

The empirical Bayes selection of $\hat{\lambda}$ selects the best hyper-hyperparameter and reintroduces maximization in a way that makes use of the intermediate posteriors during tuning, as in Eq. 4. In addition, it uses hyper-hyperparameter optimization to find $\hat{\lambda}$. This depends on the choice of hyperprior. There is flexibility in this choice; we found that a nonparametric hyperprior that places a uniform distribution over the top $T < S$ samples (by value of $f_X(\eta^{(t)})$) from Eq. 4 works well in practice, and this is what we use in Section 4 with $T = \lfloor \frac{S}{10} \rfloor$. This choice of hyperprior avoids converging on a point mass in the limit of infinite sized data $X$ and forces the approximate marginal to spread probability

Table 3: Predictive log lik. for deep latent Gaussian model (DLGM), Labeled Faces in the Wild

| Method | | Predictive Log Lik. (% Improvement on BayesOpt) |
|---|---|---|
| BayesOpt | with validation | -17071 (0.00%) |
| | without validation | -15970 (+6.45%) |
| EB-Hyp | with validation | -16375 (+4.08%) |
| | without validation | **-15872 (+7.02%)** |
| Random | | -17271 (-1.17%) |

mass across a well-performing set of models, any one of which is likely to dominate the prediction for any given data point (though, importantly, it will not always be the same model).

After the Markov chain in Eq. 5 converges, the samples $\{\eta^{(s)} \mid s = 1, \ldots, S\}$ and the (approximated) posteriors $p(\theta \mid X, \eta^{(s)})$ can be used in Eq. 4. The EB-Hyp algorithm is summarized in Algorithm 1. The dominating computational cost comes from running inference to evaluate $\mathcal{A}(X_{\text{train}}, \eta^{(s)})$. All the other steps combined are negligible in comparison.

## 4 Experiments

We apply EB-Hyp and BayesOpt to two approximate inference algorithms and data sets. We also apply uniform random search, which is known to outperform a grid or manual search (Bergstra and Bengio, 2012).

In the first experiment, we consider stochastic variational inference on latent Dirichlet allocation (SVI-LDA) applied to the 20 Newsgroups data.[3] In the second, a deep latent Gaussian model (DLGM) on the Labeled Faces in the Wild data set (Huang *et al.*, 2007). We find that EB-Hyp outperforms BayesOpt and random search as measured by predictive likelihood.

For the performance model, we use the log Gaussian process in our experiments implemented in the `GPy` package (GPy, 2012). The performance model uses the Matérn 32 kernel to express the assumption that nearby hyperparameters typically perform similarly; but this kernel has the advantage of being less smooth than the squared exponential, making it more suitable to capture abrupt changes in the marginal likelihood (Stein, 1999). Between each hyperparameter sample, we optimize the kernel parameters and the independent noise distribution for the observations so far by maximizing the marginal likelihood of the Gaussian process.

Throughout, we randomly split the data into training, validation, and test sets. To assess the necessity of a separate validation set we consider two scenarios: (1) training and validating on the train+validation data, (2) training on the train data and validating on the validation data. In either case, the test data is used only at the final step to report overall performance.

### 4.1 Latent Dirichlet Allocation

Latent Dirichlet allocation (LDA) is an unsupervised model that finds topic structure in a set of text documents expressed as $K$ word distributions (one per topic) and $D$ topic distributions (one per document). We apply stochastic variational inference to LDA (Hoffman *et al.*, 2013), a method that approximates the posterior over parameters $p(\theta \mid X, \eta)$ in Eq. 4 with variational distribution $q(\theta \mid v, \eta)$. The algorithm minimizes the KL divergence between $q$ and $p$ by adjusting the variational parameters.

We explored four hyperparameters of SVI-LDA in the experiments: $K \in [50, 200]$, the number of topics; $\log(\alpha) \in [-5, 0]$, the hyperparameter to the Dirichlet document-topic prior; $\log(\eta) \in [-5, 0]$, the hyperparameter to the Dirichlet topic-word distribution prior; $\kappa \in [0.5, 0.9]$, the decay parameter to the learning rate $(t_0 + t)^{-\kappa}$, where $t_0$ was fixed at 10 for this experiment. Several other hyperparameters are required and were kept fixed during the experiment. The minibatch size was fixed at 100 documents and the vocabulary was selected from the top 1,000 words, excluding stop words, words that appear in over 95% of documents, and words that appear in only one document.

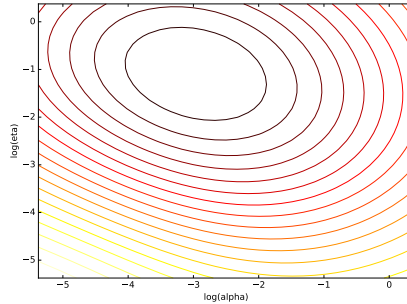

Figure 2: A 2D slice of the performance model posterior after a run of EB-Hyp on LDA. The two hyperparameters control the sparsity of the Dirichlet priors. The plot indicates a negative relationship between them.

The 11,314 resulting documents were randomly split 80%-10%-10% into training, validation, and test sets.

Table 2 shows performance in log likelihood on the test data of the two approaches. The percentage change over the BayesOpt benchmark is reported in parentheses. EB-Hyp performs significantly better than BayesOpt in this problem. To understand why, Figure 1 examines the error (negative log likelihood) on both the validation and test data for all the hyperparameters selected during BayesOpt. In the test scenario, BayesOpt chooses the hyperparameters corresponding to the left-most bar in Figure 1b because those hyperparameters minimized error on the validation set. However, Figure 1b shows that other hyperparameter settings outperform this selection when testing. For finite validation data, there is no way of knowing how the optimal hyperparameter will behave on test data before seeing it, motivating an averaging approach like EB-Hyp. In addition, Table 2 shows that a separate validation data set is not necessary with EB-Hyp. In contrast, BayesOpt does need separate validation and overfits the training data without it.

Figure 2 shows a slice of the posterior mean function of the performance model for two of the hyperparameters, $\alpha$ and $\eta$, controlling the sparsity of the document-topics and the topic-word distributions, respectively. There is a negative relationship between the two hyperparameters, meaning that the sparser we make the topic distribution for documents, the denser we need to make the word distribution for topics to maintain the same performance (and vice versa). EB-Hyp combines several models of different degrees of sparsity in a way that respects this trade-off.

## 4.2  Supervised Deep Latent Gaussian Models

Stochastic backpropagation for deep latent Gaussian models (DLGMs) approximates the posterior of an unsupervised deep model using variational inference and stochastic gradient ascent (Rezende *et al.*, 2014). In addition to a generator network, a recognition network is introduced that amortizes inference (i.e., once trained, the recognition network finds variational parameters for new data in a closed-form expression). In this experiment, we use an extension of the DLGM with supervision (Li *et al.*, 2015) to perform label prediction on a subset of the Labeled Faces in the Wild data set (Huang *et al.*, 2007). The data consist of 1,288 images of 1,850 pixels each, split 60%-20%-20% into training, validation, and test data (respectively).

We considered 4 hyperparameters for the DLGM with a one-layered recognition model: $N_1 \in [10, 200]$, the number of hidden units in the first layer of the generative and recognition models; $N_2 \in [0, 200]$, the number of hidden units in the second layer of the generative model only (when $N_2 = 0$, only one layer is used); $\log(\kappa) \in [-5, -0.05]$, the variance of the prior of the weights in the generative model; and $\log(\rho) \in [-5, -0.05]$, the gradient ascent step size. Table 3 shows performance for the DLGM. The single best performing hyperparameters were ($N_1 = 91, N_2 = 86, \log(\kappa) = -5, \log(\rho) = -5$). We find again that, EB-Hyp outperforms all the other methods on test data. This is achieved without validation.

# 5    Conclusions

We introduced a general-purpose procedure for dealing with unknown hyperparameters that control the behaviour of machine learning models and algorithms. Our approach is based on approximately marginalizing the hyperparameters by taking a weighted average of posteriors calculated by existing inference algorithms that are time intensive. To do this, we introduced a procedure for sampling informative hyperparameters from a performance model. Our approaches are supported by an efficient algorithm. In two sets of experiments, we found this algorithm outperforms optimization and random approaches.

The arguments and evidence presented in this paper point toward a tendency of the standard optimization-based methodologies to overfit hyperparameters. Other things being equal, this tendency punishes (in reported performance on test data) methods that are more sensitive to hyperparameters compared to methods that are less sensitive. The result is a bias in the literature towards methods whose generalization performance is less sensitive to hyperparameters. Averaging approaches like EB-Hyp help reduce this bias.

## Acknowledgments

Many thanks to Scott Linderman, Samantha Hansen, Eric Humphrey, Ching-Wei Chen, and the reviewers of the workshop on Advances in Approximate Bayesian Inference (2016) for their insightful comments and feedback.

## Footnotes

[1]this approach could also be called *type-III* maximum likelihood because it involves marginalizing over model parameters $\theta$, hyperparameters $\eta$, and maximizing hyper-hyperparameters $\lambda$.

[2]$Z$ can be easily calculated if $\eta$ is discrete or if $p(\eta)$ is conjugate to $p(X \mid \eta)$. In non-conjugate continuous cases, $\eta$ may be discretized to a high granularity. Since EB-Hyp is an active procedure, the limiting computational bottleneck is to calculate the posterior of the performance model. For GPs, this is an $\mathcal{O}(S^3)$ operation in the number of hyperparameter evaluations $S$. If onerous, the operation is amenable to well established fast approximations, e.g,. the inducing points method (Hensman *et al.*, 2013).

[3]`http://qwone.com/~jason/20Newsgroups/`

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
