[Reviews · NeurIPS 2017]

Reviewer 1



This paper describes an alternative to Bayesian Optimization (BO). In BO, only the hyperparameter setting corresponding to the “best” model (evaluated on validation data) is kept and all others are discarded. The authors argue that this is wasteful, given the cost of training and evaluating several models. As a solution, they proposed an empirical-Bayes-inspired approach consisting in averaging hyperparameters across Bayesian optimization evaluations. The paper is well written and the contribution is clear. Indeed, keeping only the “best performing” evaluation is wasteful and discards most of the uncertainty associated with the stochasticity arising from subsampling the training set to generate a validation set. The main benefit of this method seems to be the ability to work without a validation set. In fact, the method works better when no validation set is used. The authors seem to give two reasons for this: 1) not holding out a validation set gives more data to train on 2) evaluating on a (necessarily small) validation set introduces variance. Although this is an interesting paper, the comparison with Bayesian optimization methods is not as exhaustive as it should be. For instance, to see if the improvement over BO methods is arising from point (1) above, the authors could take the best candidate from BO and simply retrain on validation+training data (concatenated). Another obvious comparison that is still missing is to simply average the prediction of all settings considered during a BO run. Minor: “validation” is misspelled in Figure 1

Reviewer 2



# Update after author feedback I maintain my assessment of the paper. I strongly urge the authors to improve the paper on the basis of the reviewers' input; this could turn out to become a very nice paper. Specially important IMHO - retrain on validation+train (will come as an obvious counter-argument) - clarify the use of $\lambda$; R2 and R3 got confused - variance in experiments table 2+3 (again, would be an easy target for critique once published) # Summary of paper Bayesian optimisation (BO) discards all but the best performance evaluations over hyperparameter sets; this exposes to overfitting the validation set, and seems wasteful. The proposed method "averages" over choices hyperparameters $\eta$ in a Bayesian way, by treating them as samples of a posterior and integrating over them. This requires introducing an extra layer of hyper-hyperparameters (introducing $\lambda$, subjected to empirical Bayes optimisation). An algorithmic choice is required to define an "acquisition strategy" in BO terms, ie a model for $r^{(s)}$, as well as a model for hyperprior $p(\eta | \lambda)$. The proposed method does not require a separate validation set. # Evaluation The paper is very clearly written, without any language issues. It represents the literature and the context of the methods well. The problem statement and resolution process is clear, as is the description of the adopted solutions and approximations. The authors take care in preventing misunderstandings in several places. The discussion of convergence is welcome. The experiments are non-trivial and help the argumentation, with the caveat of the significance of results mentioned above. The method is novel, it applies the idea of EB and hierarchical Bayes to the BO method. I believe the approach is a good, interesting idea. # Discussion Discussion lines 44-47 and 252 sqq: it may sound like the case presented here was cherry-picked to exhibit a "bad luck" situation in which the leftmost (in fig 1) hyperparameter set which is selected by BO also exhibits bad test error; and in a sense, it is indeed "bad luck", since only a minority of hyperparameter choices plotted in fig 1 seem to exhibit this issue. However, dealing with this situation by averaging over $\eta$'s seems to be the central argument in favour of the proposed EB-Hyp method. I buy the argument, but I also recognise that the evidence presented by fig 1 is weak. The following advantage does not seem to be discussed in the paper: since a separate validation set is not needed, the training set can be made larger. Does this explain the observed performance increase? An analytical experiment could help answer this. Does the comparison between BO with/without validation in tables 2 and 3 help? In algo 1, I am not clear about the dependence on $\hat{\lambda}$. I am assuming it is used in the line "return approximation to ..." (I suggest you write $p(X^*|X_\textrm{train}, \hat{\lambda})$ to clarify the dependence.) I assume that in line "find optimal...", we actually use, per lines 200-208, $\hat{\lambda} = \arg \max \frac{1}{T} \sum_s p(\eta ^{(s)} | \lambda) f^{(s)}_X $. This is worth clarifying. But then, I'm confused what $p(\eta)$ in line "calculate hyperparameter posterior..." and eq 5 is? Is this an input to the algorithm? EB-Hyp seems to carry the inconvenient that we must consider, as the "metric of interest" in BO terms, the predictive likelihood $f^{(s)}_X$, while BO can use other metrics to carry out its optimisation. This restriction should be discussed. ## Experiments How many steps/ evaluations of $A$ do we use in either experiment? We are using the same number for all configurations, I suppose? How does the performance increase with the number of steps? Does EB-Hyp overtake only after several steps, or is its performance above other methods from the start? I find it hard to assign significance to the improvements presented in table 2 and 3, for lack of a sense of performance variance. The improvements are treated as statistically significant, but eg are they immune against different random data splits ? As it stands, I am not convinced that the performance improvement is real. # Suggestions for improvement, typos ## Major Fig 2 is incomplete: it is missing a contour colour scale. In the caption, what is a "negative relationship"? Please clarify. What is the point of fig 2 ? I can't see where it is discussed? ## Minor fig 1a caption: valiation fig 1: make scales equal in a vs b algo 1: increment counter $s$. Number lines. line 228: into into line 239: hyerparameters

Reviewer 3



This is a great paper demonstrating the importance of *integrating out* hyperparameters instead of optimizing them. I liked the point of not needing a separate validation set and appreciate the risk of overfitting to the validation set when using BayesOpt. These are all mistakes practitioners of deep learning make regularly, including myself. I also really liked the idea of using Thomson sampling instead of a closed for acquisition function in order to generate better samples while performing the integral. Obviously sampling from the prior would be much more inefficient, and I like how Thomson sampling unites Bayesian optimization with efficient Monte Carlo techniques. Now for the criticisms: 1) It is quite unfortunate that the results with fixed \lambda are not competitive. Have the authors tried partial runs to get a noisier estimate of the loglik for a particular \eta? This could increase the number of samples going into the GP. 2) The optimization over \lambda in Algorithm 1 appears weird -- isn't the loop over a fixed lambda? How is this optimization performed in practice? 3) In it unfortunate that the only experiments are for unsupervised learning, you could apply this to the supervised learning setting. It would be nice to have an experiment with deep convnets or RNNs on a large dataset like ImageNet for example.